# Estimated burden, and associated factors of Urinary Incontinence among Sub-Saharan African women aged 15–100 years: A systematic review and meta-analysis

**Martin Ackah** [1] *, **Louise Ameyaw** [2], **Mohammed Gazali Salifu** [3], **Cynthia OseiYeboah** [1], **Abena Serwaa Ampomaa Agyemang** [4], **Kow Acquaah** [1], **Yaa Boatema Koranteng** [1], **Asabea Opare-Appiah** [1]

1 Department of Physiotherapy, Korle Bu Teaching Hospital, Accra, Ghana, 2 Department of Medicine, Achimota Government Hospital, Accra, Ghana, 3 Policy Planning Budgeting Monitoring and Evaluation Directorate, Ministry of Health, Accra, Ghana, 4 Department of Physiotherapy, Greater Accra Regional Hospital, Accra, Ghana

\* martinackah10@gmail.com

## Abstract

Hospital and community based-studies had been conducted for Urinary Incontinence (UI) in Sub-Sahara Africa (SSA) countries. A significant limitation of these studies is likely under-estimation of the burden of UI in SSA. It is therefore, imperative that a well-structured systematic review and meta-analytical models in SSA are required to accurately and reliably estimate the burden of UI. Medline/PubMed, Google Scholar, Africa Journal Online (AJOL) were searched to identified data on burden of UI studies in SSA. Two independent authors performed the initial screening of studies based on the details found in their titles and abstracts. The quality of the retrieved studies was assessed using the Newcastle-Ottawa Quality Assessment instrument. The pooled burden of UI was calculated using a weighted inverse variance random-effects model. A sub-group and meta-regression analyses were performed. Publication bias was checked by the funnel plot and Egger's test. Of the 25 studies included, 14 were hospital-based, 10 community- based, and 1 university-based studies involving an overall 17863 participants from SSA. The systematic review showed that the prevalence of UI ranged from 0.6% in Sierra Leone to 42.1% in Tanzania. The estimated pooled burden of UI across all studies was 21% [95% CI: 16%-26%, $I^2$ = 91.01%]. The estimated pooled prevalence of stress UI was 52% [95% CI: 42%-62%], urgency UI 21% [95% CI: 15%-26%], and mixed UI 27% [95% CI: 20%-35%]. The common significant independent factors were; parity, constipation, overweight/obese, vaginal delivery, chronic cough, gestational age, and aging. One out of every five women in SSA suffers from UI. Parity, constipation, overweight/obesity, vaginal delivery, chronic cough, gestational age, and age were the most important risk variables. As a result, interventions aimed at reducing the burden of UI in SSA women aged 15 to 100 years old in the context of identified determinants could have significant public health implications.

**Data Availability Statement:** All relevant data are within the manuscript.

**Funding:** The authors received no specific funding for this work.

**Competing interests:** The authors have declared that no competing interests exist.

## Introduction

Pelvic Floor Disorders (PFD) affects millions of women worldwide [1–3]. About 10% of women have surgery for Urinary Incontinence (UI), pelvic organ prolapse, or both, according to a regional survey in the United States, and 30% of those women have two or more surgical procedures in their lifetime [3, 4]. Wu and colleagues estimated that 25% of women in affluent countries suffers from one or more PFDs [5] with UI being the most common [6].

The International Continence Society (ICS) defines UI as the involuntary leakage of urine, with three basic subtypes identified: urgency UI (UUI), stress UI (SUI), and mixed UI (MUI; both UUI and SUI) [7, 8]. It is a widespread problem with an estimated global burden of nearly 5.0% to 55% with detrimental consequences on social life, personal relationships, feelings, sleep, and vitality [9, 10]. A comprehensive review and meta-analysis of 54 studies comprising 138722 women aged 10 to 90 years in Low- and Middle-Income Countries estimated the burden of UI to be 26% [1]. In addition, the prevalence of UI ranged from 2.8% in Nigeria to 57.7% in the Islamic Republic of Iran [11]. UI is frequently underestimated and underdiagnose in developing and industrialized countries [11].

In comparison to patients with continence, recent investigations have shown that UI is a predictor of death [12, 13]. As a results, in order to strengthen continence programs, health systems should be able to estimate the burden especially in a region with a frail health system, such as SSA. Hospital and community based-studies had been conducted for UI in SSA countries [14–17]. A significant limitation of these observational studies is likely under-estimation of the burden of UI in SSA [16]. Following a thorough search of the literature, it was revealed that no prior systematic review and meta-analysis addressing the burden of UI and associated factors on the African continent has yet been conducted and published.

Many risk factors, such as multiple pregnancies, positive family history, parity, episiotomy use, body mass index, advanced age, spontaneous perineal tear at delivery and so on, appear to be implicated for UIs in High Income Countries (HIC) [2, 18–20].There are insufficient researches to draw conclusion on the risk factors for UI in SSA. In addition, large, diversified population-based studies have assessed prevalence rates of UI; however, there is limited robust evidence describing the burden of UI amongst women in SSA, where parity on average is higher than those in High-Income Countries [21].

It is therefore, imperative that a well-structured systematic review and meta-analytical models in SSA are required to accurately and reliably estimate the burden, and associated factors for UI. In this context, the current study aims to assess the burden of UI in SSA, as well as the risk factors associated with it. Thus, the review sought to answer the questions; what is the burden of UI in SSA? and what are the factors associated with the burden of UI in SSA?

## Methods

### Overview

This systematic review was registered in PROSPERO [CRD42021267551]. This systematic review and meta-analysis was conducted and reported according to the guidelines of the Preferred Reporting Items for Systematic Review and Meta-Analysis (PRISMA) [22] [S1 Checklist].

### Eligibility criteria

**Inclusion criteria.** Observational studies such as longitudinal, cohort, case control and cross-sectional studies reporting prevalence/or risk factors of UI were incorporated in the current review, as well as conference abstracts with enough information to calculate prevalence

UI. Original observational studies published in English and adult SSA woman aged ≥18 years were included. Burden of UI studies that compared both SSA men and women, only information on the women were extracted. Finally, both hospital and community/population-based studies were included and later stratified in the pooled meta-analysis.

**Exclusion criteria.** Studies reporting animal studies, reviews, commentaries, letter to editors were excluded. Prevalence of UI articles published in other languages were excluded. Studies that looked at the management and treatment of UI, as well as quality of life, depression, without data on burden of UIs were excluded. Studies from North Africa countries, other LMICs, and HIC were also excluded. UI studies involving children and adolescent females were also excluded.

## Data sources and search strategy

Medline/PubMed, Google Scholar, Africa Journal Online were searched to extract data on burden of UI studies in SSA countries, as well as their respective risk factors' information. The articles that were considered were published between 2000/1/1 and 2021/9/30. There was also a manual search of the reference lists of the studies that were included. Medical Sub-Heading (MeSH) terms and free text were used in the search approach. These terms were coupled with the Boolean operators 'OR' and 'AND'. The keywords included; Burden, prevalence, Pelvic Floor Disorders, Urinary Incontinence, Sub-Saharan Africa. The final search strategy is displayed in (S1 Table).

## Selection process

To ensure a rigorous review strategy, articles were reviewed individually by two independent co-authors [MA, and KA]. The data screening was done in two stages; the title abstract screening, followed by the full-text screening. Both steps were completed independently by two review authors. A third reviewer (ASA) was available to resolved the disagreement between MA and KA. Finally, all the studies were imported into Mendeley desktop reference manager.

## Data collection and management

MA and KA independently extracted data into an excel sheet, and discrepancies were resolved through discussion. Extracted data were; Author's name, year of publication, country, age, study design, population, sample size, setting, information on the burden of UI. Finally, significant independent risk factors from the individual studies were extracted.

## Quality assessment and risk of bias

The quality of the retrieved studies were assessed using the Newcastle-Ottawa Quality Assessment instrument, which was customized for cross-sectional research [23]. The assessment's goal was to determine the research' internal and external validity, as well as to reduce the possibility of bias [23]. The findings of the quality assessment are presented in S2 Table.

## Data synthesis and strategy

The pool burden of UI was calculated using a weighted inverse variance random-effects model. This was visually represented using the forest plot. The presence of heterogeneity among studies was quantified by estimating variance using the $I^2$ statistics [24]. The $I^2$ takes values between 0 and 100%, and a value of 0% indicates absence of heterogeneity. $I^2$ was interpreted based on Higgins and Thompson classification, percentages of 25%, 50% and 75% was considered as low, moderate and high heterogeneity, respectively [24].

A subgroup analysis was performed to determine the sources of heterogeneity on the study characteristics (year of publication, sample size, setting, and sub-regions). The funnel plot and Egger's regression test was used to screened for publication bias. Finally, meta-regression was performed to assessed the factors influencing the review's heterogeneity.

## Results

### Study selection process

The study identified one thousand nine hundred and twenty-nine (1929) articles from PubMed, AJOL, and Google scholar, out of which 700 were removed as duplicate. One thousand two hundred and twenty-nine articles were screened, of which 1071 papers were excluded. A total of 148 publications were evaluated for eligibility, with 25 studies [14–17, 25–45] being included in the current review [Fig 1]. This study included a total of 17863 women from SSA.

### Characteristics of the included studies and quality assessment

The characteristics of the included studies are shown in Table 1. The studies were published between 2005 and 2021. Nigeria had the highest number of eligible studies [n = 14], followed by Ethiopia [n = 3], followed by Ghana, and South Africa with two studies each. The age of the participants ranged from 15 to 100 years. Of the 25 studies, 14 were hospital-based, 10 community- based, and 1 university-based studies. The sample size ranged from 100 to 5000, with overall 17863 participants from SSA. Seventy-two percent [n = 17] of the articles had low risk of bias.

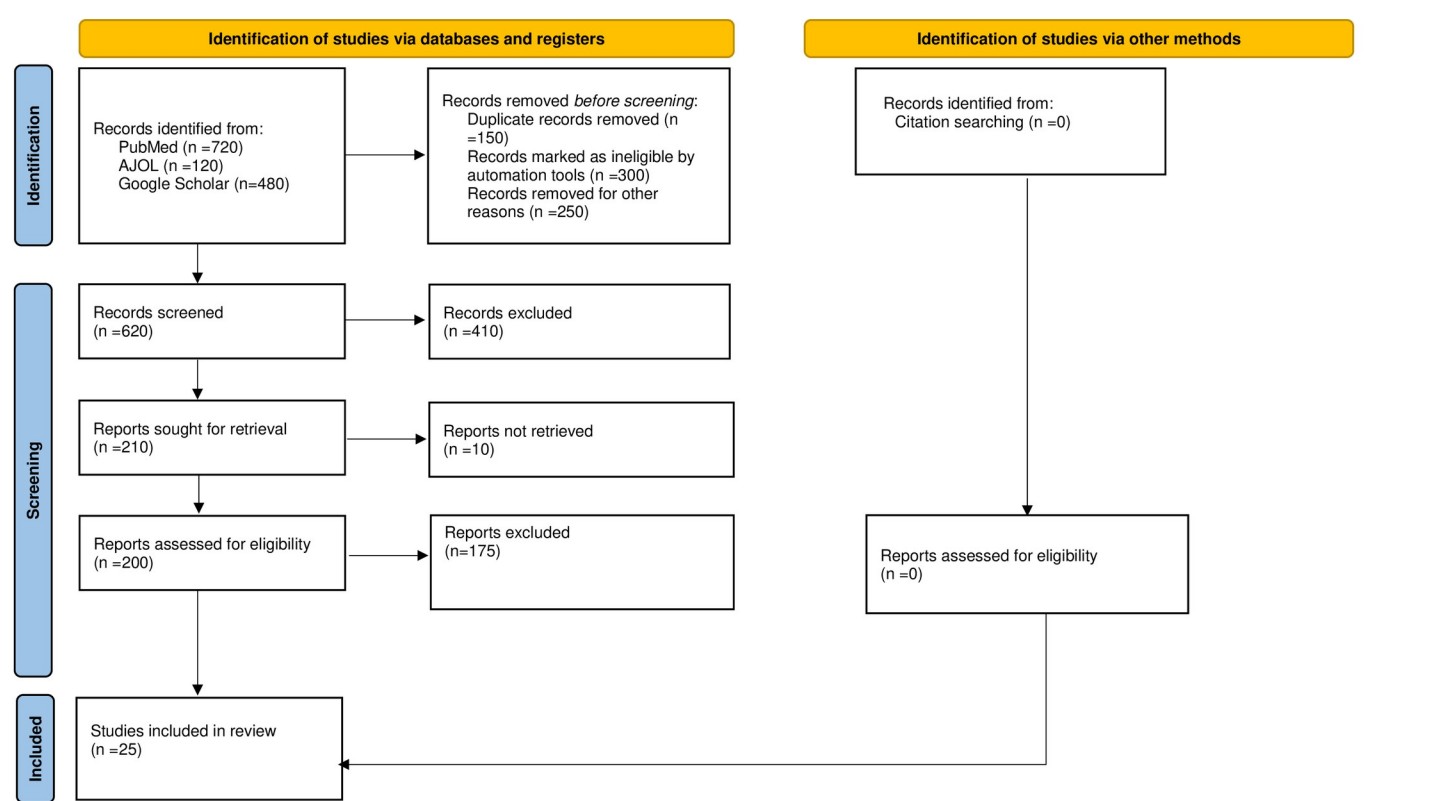

**Fig 1. PRISMA 2020 flow diagram for new systematic reviews which included searches of databases, registers and other sources.**

                    

**Table 1. Characteristics of the included Sub-Saharan African women studies on urinary incontinence.**

| SN | Study ID | Year of publication | Country | Study Design | Age/year | Sample Size | Burden [%] | Stress UI [%] | Urgency UI [%] | Mixed UI [%] | Setting | Risk of bias assessment |
|---|---|---|---|---|---|---|---|---|---|---|---|---|
| 1 | Berhe et al. [14] | 2020 | Ethiopia | Cross-sectional | 18–45 | 317 | 4.3 | 58.9 | 10.9 | 30.1 | Hospital | Low |
| 2 | Demissie et al. [15] | 2021 | Ethiopia | Cross-sectional | 19–70 | 542 | 3.3 | | | | Community | Low |
| 3 | Ofori et al. [16] | 2020 | Ghana | Cross-sectional | 19–88 | 400 | 12 | 22.9 | 33.3 | 20.8 | Hospital | Low |
| 4 | Adanu et al. [42] | 2005 | Ghana | Not reported | 17–70 | 200 | 42.0 | 100.0 | 0.0 | 0.0 | Hospital | Low |
| 5 | Balde et al. [44] | 2020 | Guinea | Retrospective Cohort | 15–70 | 1770 | 10.2 | | | | Hospital | Low |
| 6 | Bekele et al. [45] | 2016 | Ethiopia | Cross-sectional | 16–40 | 422 | 11.4 | | | | Hospital | Low |
| 7 | Bowling et al. [25] | 2010 | Liberia | Not reported | Not reported | 424 | 1.7 | | | | Community | Moderate |
| 8 | Ojengbede et al. [34] | 2010 | Nigeria | Prospective Cohort | 15–45+ | 5001 | 2.8 | | | | Community | Low |
| 9 | Usifoh et al. [38] | 2012 | Nigeria | Cross-sectional | 15–60+ | 412 | 29.4 | 44.6 | 14.9 | 40.5 | Community | Low |
| 10 | Rabiu et al. [29] | 2015 | Nigeria | Cross-sectional | 15–44 | 257 | 15.2 | 43.6 | 46.2 | 20.2 | Hospital | Moderate |
| 11 | Okunola et al. [35] | 2018 | Nigeria | Cross-sectional | 18–45 | 442 | 28.1 | 62.1 | 24.2 | 19.4 | Hospital | Low |
| 12 | Abiola et al. [40] | 2016 | Nigeria | Cross-sectional | Not reported | 229 | 12.7 | 58.6 | 27.6 | 17.2 | Community | Low |
| 13 | Akinlusi et al. [17] | 2020 | Nigeria | Cross-sectional | 25–67 | 395 | 32.9 | 54.6 | 23.1 | 22.3 | Hospital | Low |
| 14 | Adaji et al. [41] | 2009 | Nigeria | Cross-sectional | 15–42 | 204 | 21.1 | 60.4 | 25.6 | 9.3 | Hospital | Moderate |
| 15 | Yağmur et al. [39] | 2021 | Nigeria | Cross-sectional | 40–69 | 286 | 30.1 | 30.2 | 7.0 | 31.4 | Community | Moderate |
| 16 | Badejoko et al. [43] | 2015 | Nigeria | Cross-sectional | 20–100 | 1250 | 5.2 | 35.4 | 46.2 | 18.6 | Hospital | Low |
| 17 | Bello et al. [36] | 2018 | Nigeria | Cross-sectional | 16–46+ | 500 | 21.4 | 40.2 | 8.4 | 51.4 | Hospital | Low |
| 18 | Njoku et al. [32] | 2020 | Nigeria | Cross-sectional | Not reported | 658 | 16.1 | 73.6 | 16.9 | 9.4 | Hospital | Low |
| 19 | Irshad et al. [28] | 2021 | Nigeria | Cross-sectional | 15–45 | 282 | 26.2 | 56.8 | 8.1 | 33.8 | Hospital | Moderate |
| 20 | Obioha et al. [33] | 2015 | Nigeria | Prospective Cohort | Not reported | 230 | 12.2 | | | | Hospital | Moderate |
| 21 | Gashugi et al. [26] | 2005 | Rwanda | Not reported | 20–64 | 1030 | 41.9 | | | | Community | Low |
| 22 | Patel et al. [27] | 2014 | Sierra Leone | Not reported | Not reported | 1320 | 0.6 | | | | Community | Moderate |
| 23 | Skaal et al. [37] | 2011 | South Africa | Not reported | Not reported | 145 | 31.7 | | | | University | Low |
| 24 | Madombwe et al. [30] | 2010 | South Africa | Not reported | 21–76 | 100 | 35.4 | | | | Community | Low |
| 25 | Masenga et al. [31] | 2019 | Tanzania | Cross-sectional | 18–90 | 1048 | 42.1 | 39.0 | 22.0 | 39.0 | Community | Low |

SN = Serial Number, UI = Urinary Incontinence

                    

## Estimated Burden of Urinary Incontinence amongst Sub-Saharan Africa women

The systematic review showed that the prevalence of UI ranged from 0.6% in Sierra Leone to 42.1% in Tanzania. In the meta-analysis, the pooled estimate of the burden of UI across all studies was 21% [95% CI: 16%-26%, $I^2$ = 91.01%]. A substantial statistically significant heterogeneity was detected across the studies [Fig 2].

With respect to the sub-types of UI, 14 studies were included, and the estimated pooled prevalence of stress UI was 52% [95% CI: 42%-62%, $I^2$ = 70.78%], urgency UI 21% [95% CI: 15%-26%, $I^2$ = 0.00%], and mixed UI 27% [95% CI: 20%-35%, $I^2$ = 46.37%] [Table 2].

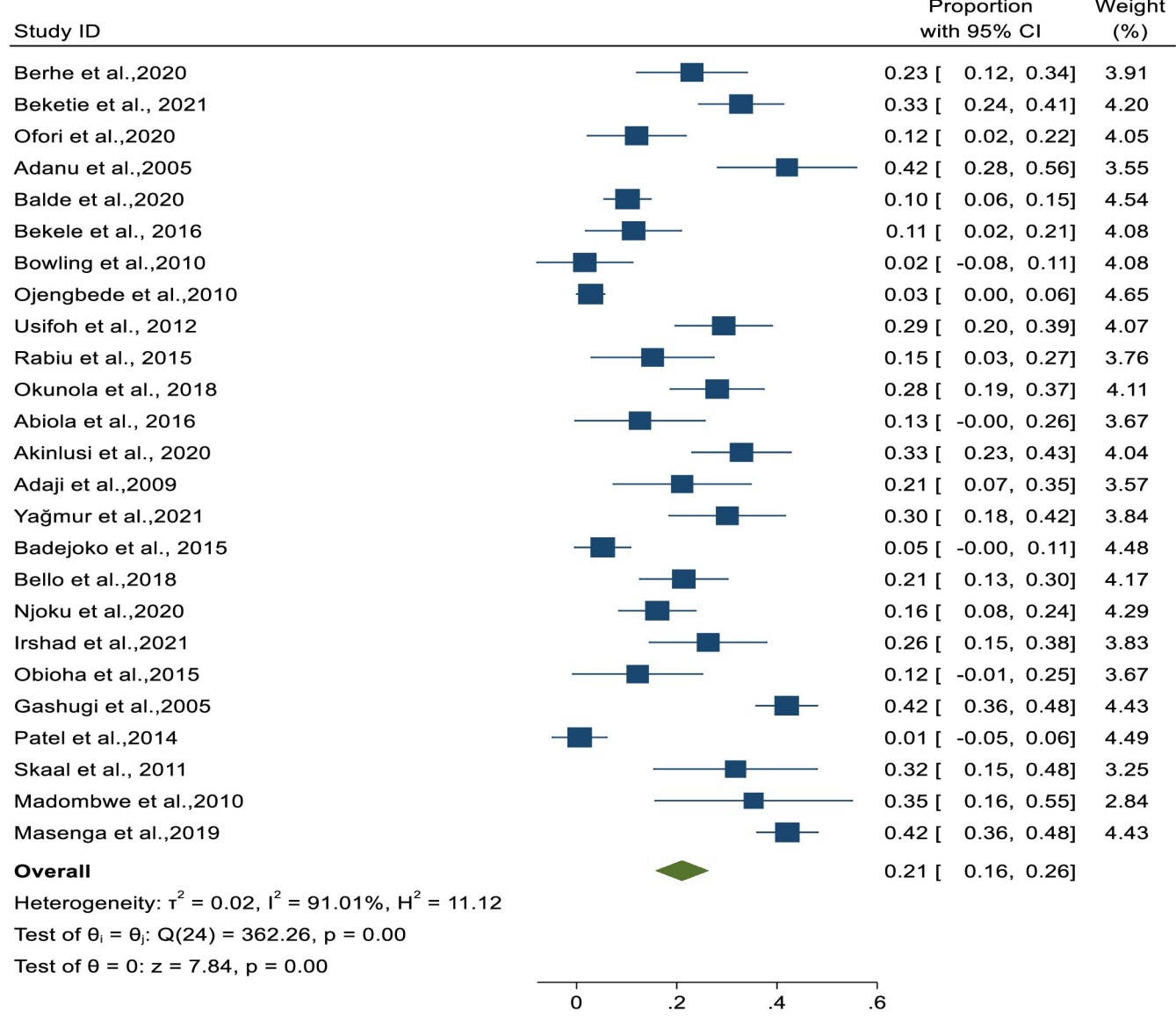

**Fig 2. Forest plot of pooled Burden of Urinary Incontinence amongst SSA women.**

**Table 2. Sub-group analysis of types of UI, sub-region, and setting of study.**

| Sub group | Variable | Dataset | Pooled burden % [95%CI] |
|---|---|---|---|
| **Sample size** | | | |
| | ≤400 | 12 | 24.0 [18.0–30] |
| | >400 | 13 | 19.0 [10.0–27] |
| **Types of UI** | | | |
| | Stress UI | 14 | 52.0 [42.0–62.0] |
| | Urgency UI | 14 | 21.0 [15.0–26.0] |
| | Mixed UI | 14 | 27.0 [20.0–35.0] |
| **Sub-Region** | | | |
| | East Africa | 5 | 31.0 [19.0–42.0] |
| | West Africa | 18 | 16.0 [11.0–21.0] |
| | South Africa | 2 | 23.0 [12.0–33.0] |
| **Study setting** | | | |
| | Hospital based | 14 | 18.0 [13.0–22.0] |
| | Community-based | 10 | 23.0 [16.0–26.0] |

## Sub-group analysis

Sub-group analysis were performed with regards to sub-region [East africa vs.West Africa vs. South Africa], and setting [Hospital-based vs. Community-based studies]. The pooled estimate of UI in East Africa was 31% [95% CI: 19%-42%], West Africa was 16% [95% CI: 11%-21%], and South Africa 35% [95% CI: 15%-55%]. With regards to settings, the estimated burden of UI was 18% [95% CI: 13%-22%] for hospital based studies, and 23% [95% CI: 12%-33%] for community-based studies. The findings are summarised in Table 2.

## Meta-regression

Meta-regression revealed non-sigifcant, decreased trend in the year of publication (coefficient = -0.0033, p = 0.533), and significant decreased trend in the number of sample size (coefficient = -0.0001, p = 0.048) with increasing burden of UI amongst women in SSA [S3 Table].

## Publication bias

There was no evidence of publication bias in both the subjective funnel plot [Fig 3] and the objective Egger's regression test (z = 1.74, p = 0.0825).

## Systematic review of associated factors for UI amongst Sub-Saharan African women

Systematic review of factors associated with UI amongst SSA women is shown Table 3. The significant independent factors were; parity [14, 31, 32], constipation [14, 17, 45], overweight/ obese [17, 35, 45], vaginal delivery [32, 34, 35] chronic cough [16, 45], gestational age [14, 35] and aging [16, 32]. For example, Berhe et al. found that multiparous women had approximately 6 times higher chances of developing UI compared to primigravida women in Ethiopia [14]. A population-based study in rural Tanzania showed that women who had experienced multiple births had 2 times chance of reporting UI [31]. Also, women with history of constipation had 2times chance of reporting UI compared to women without prior constipation in a Nigerian study [17]. Similarly, an Ethiopian study reported history of constipation as an associated factor for UI [45]. The results are presented in Table 3.

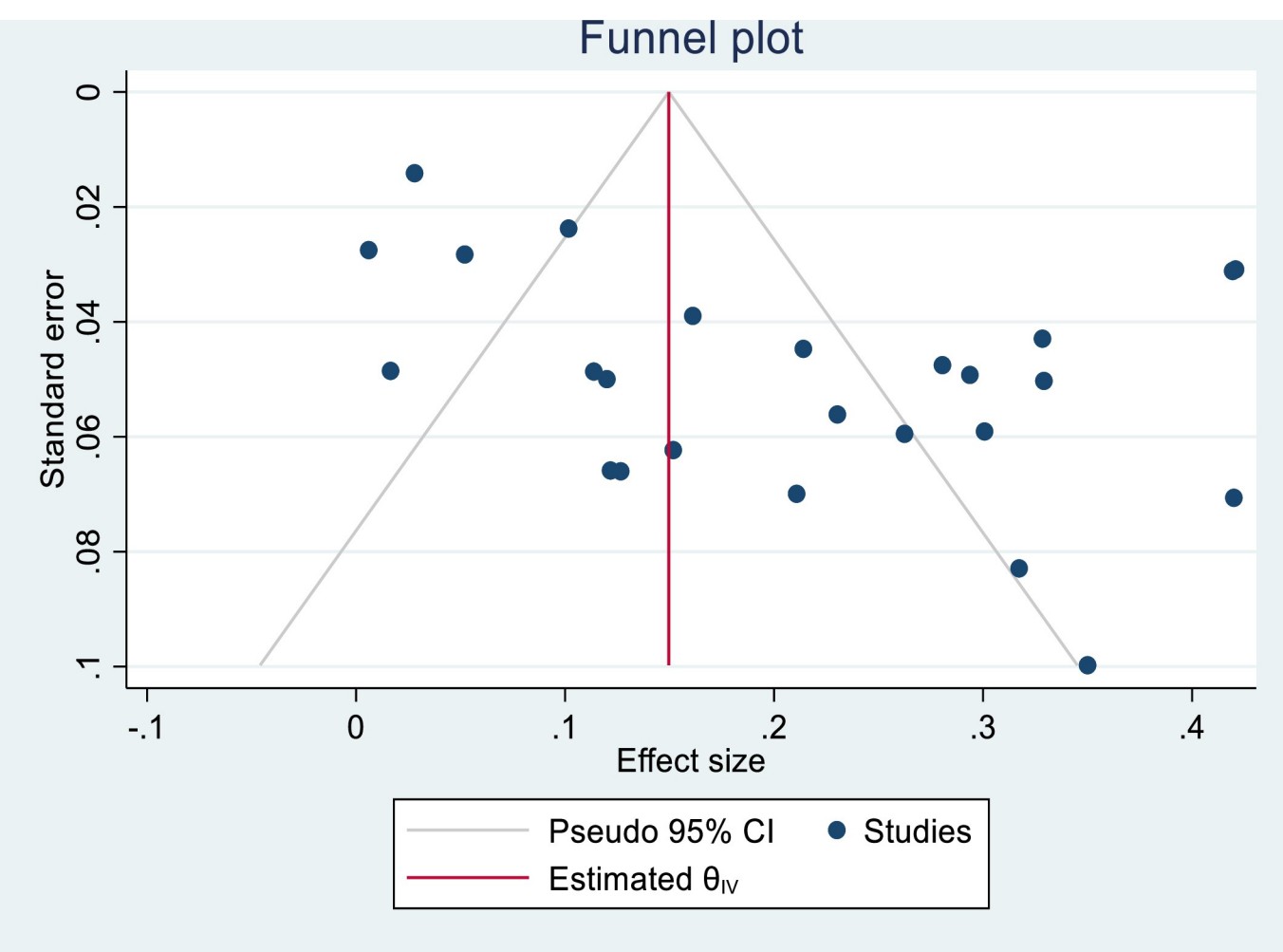

**Fig 3. Assessment of publication bias using the funnel plot.**

## Discussion

The systematic review and meta-analysis included 25 studies with 17863 participants from 9 countries across SSA. We present a comprehensive review of the burden of UI and associated variables in SSA women in this study. The systematic review showed that the burden of UI ranged from 0.6% in Sierra Leone to 42.1% in Tanzania. According to the meta-analysis, the sub-region has an estimated pooled burden of UI to be 21% [95% CI: 16%-26%]. The study further revealed that the commonest type of UI was stress UI (52%), mixed UI (27%), and urgency UI (21%). There were a wide range of prevalent estimates among the participants. For example, Patel et al. [27] discovered that less than 1% of Sierra Leonean women have UI, but Masenga et al. [31] found that 42.1% of Tanzanian women have UI. The current estimated burden is lower than reported by Xue et al in China [46], Mostafaei et al in developing countries [11], and Batmani et al in the global estimate [47]. Similarly, a study in the United States found that 45% of participants aged 30–60 years reported UI [48], which is significantly higher than the current estimate. The comparatively lower burden in the current review could be attributed to variation in the method used, under-reporting, underdiagnosis as a results of low health seeking behavior among SSA women [49]. As a result, initiatives to reduce the burden

**Table 3. Systematic review of associated factors for UI amongst Sub-Saharan African women.**

| SN | Study ID | Country of Study | Type of Study | Independent Risk Factors | p-value |
|---|---|---|---|---|---|
| 1 | Berhe et al. [14] | Ethiopia | Cross-sectional | Gestational age | ** |
| | | | | Parity | ** |
| | | | | Prior Miscarriage | ** |
| | | | | Constipation | ** |
| | | | | Respiratory Problem | ** |
| | | | | Weak PFM | ** |
| 2 | Ofori et al. [16] | Ghana | Cross-sectional | Age>60 | ** |
| | | | | History of chronic cough | ** |
| 3 | Bekele et al. [45] | Ethiopia | Cross-sectional | Episiotomy | ** |
| | | | | Constipation | ** |
| | | | | Obese women | ** |
| | | | | Chronic cough/Sneezing | ** |
| | | | | Asthma/Allergies/Sinusitis | ** |
| 4 | Okunola et al. [35] | Nigeria | Cross-sectional | overweight/obese | ** |
| | | | | Gestational age | ** |
| | | | | Previous Vaginal/Instrumental delivery | ** |
| 5 | Akinlusi et al. [17] | Nigeria | Cross-sectional | Previous Constipation | ** |
| | | | | overweight/obese | ** |
| 6 | Njoku et al. [32] | Nigeria | Cross-sectional | Age>40 | ** |
| | | | | Parity>3 | ** |
| | | | | Low educational level | ** |
| | | | | Vaginal/Instrumental Delivery | ** |
| | | | | Carry Heavy Load | ** |
| | | | | Farming | ** |
| 7 | Masenga et al. [31] | Tanzania | Cross-sectional | Parity | ** |
| | | | | Delivery at home | ** |
| | | | | Labour>24hrs | ** |
| 8 | Ojengbede et al. [34] | Nigeria | Prospective Cohort | Vaginal Delivery | ** |
| | | | | Diabetes | ** |

**Significant at p<0.05

of UI amongst SSA women aged 15–100 years could have significant public health implications.

The pooled estimate of UI in East Africa was 31%, West Africa was 16%, and South Africa 35%. Tanzania [31] reported the greatest prevalence in East Africa, while Ethiopia [14] recorded the lowest. Ofori et al. [16] found the highest prevalence of UI in West Africa, while Patel et al. [27] found the lowest. In the southern SSA, both the highest and lowest burden were reported from South Africa [30, 37]. The sub-regional variance could be attributed to the sample size included from individual countries. For example, a trend meta-regression analysis in this review found an inverse relationship between the burden of UI and sampled size.

In addition, our stratified analysis based on study setting showed that community dwelling women in SSA had an estimated burden of 23% compared to hospital-based of 18%. Ten and fourteen studies reported on community and hospital-based studies respectively.

Three studies reported that parity was independently associated with UI [14, 31, 32]. Masenga et al. [31] adduced that women with at least three children have a two-fold increased risk of contracting UI in Tanzania. Similarly, Njoku et al. [32] in Nigeria corroborated with

this findings. In Ethiopia, multiparous women had a 6 times higher risk of UI than primigravida women [14]. The current findings are supported meta-analyses [46, 47, 50]. The findings are ascribed to pelvic floor musculature and connective tissue injury that affects normal urine continence function during parturition. [50, 51].

Constipation was another commonly reported independent factors associated with UI in SSA [14, 17, 45]. The finding is consistent with systematic reviews conducted in China [46] and worldwide [47]. The mechanism underpinning this association is not well understood, although in an animal research, Chen et al. found that colon distension increased contractility, similar to bladder distention and the vesicovascular reflex, hence it could be inferred that chronic colorectal distension caused by constipation or chronically high abdominal muscle pressure during defecation limits bladder distension which exacerbates irritative bladder symptoms [52, 53].

Another important factor associated with UI among women in SSA was overweight/obese Three studies [17, 35, 45] reported on this factor. For instance, a study in Nigeria reported that an overweight/obese women had 60% increased odds of UI compared to apparently normal body weight women. Bekele et al. [45] reported similar magnitude of the event. The current result is in accordance with Batman et al. [47]. Excess body weight is thought to increase abdominal pressure, which raises bladder pressure and urethral mobility, causing stress UI and worsening detrusor instability and overactive bladder [54]. In the same vein, obesity/overweight can create chronic strain on the soft tissues, and other pelvic floor structures, hence, straining and weakening these important urethral mobility systems [55].

Furthermore, vaginal delivery [32, 34, 35], chronic cough [16, 45], and gestational age [14, 35] were all found to be factors associated with UI amongst SSA women aged 15–100 years. This is consistent with several observational studies and systematic reviews [46, 47, 50].

Finally, two studies identified aging as a risk factor for UI [16, 32]. In Ghana, women aged 60 were approximately three times more likely than women aged 18–39 to have urinary incontinence [42]. In Nigerian study, Njoku et al. [32] estimated that women over the age of 40 had a five-fold increased risk of urine incontinence.

Our findings should be viewed in the context of some caveats. First, there was significant heterogeneity among the studies. Second, studies from Central Africa were scarce, thus no research from this region was included which could affect generalization of our findings. Furthermore, only articles published in English were considered. Finally, the authors identified relevant studies using a small database. However, heterogeneity is common in meta-analyses of observational data, and it does not always invalidate the conclusions. This is the first and largest systematic review and meta-analysis on the burden of UI among SSA women. Our results are more reliable evident by no obvious publication bias.

## Recommendations, research and policy implications

Our research showed that, at least 21% of women in SSA have some form of urinary incontinence. This implies that there is the need to increase public health education and create awareness among people in order to promote health seeking behaviors among women. Healthy practices can be encouraged among women in order to reduce the disease burden in women. Such practices may include; maintaining a healthy body weight through exercise and diet to reduce the incidence of obesity and its complications. Avoiding constipation by taking in a high fiber diet, adequate intake of water, reducing immobility and regular emptying of bowel. Women should be taught how to strengthen their pelvic floor muscles especially during and after pregnancy by performing kegel's exercises. Family planning methods should be made accessible to all irrespective of their social and economic status so women can effectively

exercise their rights to choose the number of children they would want to have while taking into consideration their health. Finally, there are few studies examining the burden of UI, particularly in SSA's eastern and southern regions. It is therefore critical that more researches be undertaken in these sub-regions in order to obtain a complete, accurate, and consistent picture of this treatable condition.

## Conclusion

This is the first and most comprehensive systematic review and meta-analysis on the burden of UI amongst SSA women aged 15 to 100 years. According to the study, one out of every five women in SSA suffers from UI. Parity, constipation, overweight/obesity, vaginal delivery, chronic cough, gestational age, and age were the most important risk variables. As a result, interventions aimed at reducing the burden of UI in SSA women aged 15 to 100 years old in the context of identified determinants could have significant public health implications.

## Supporting information

**S1 Checklist. PRISMA checklist.**
(DOCX)

**S1 Table. Search strategies for the individual database.**
(DOCX)

**S2 Table. Quality assessment.**
(DOCX)

**S3 Table. Meta-regression of burden of UI in SSA women by sample size and year of publication.**
(DOCX)

## Acknowledgments

We would like to express our gratitude to all who contributed to the writing of the reviewed articles in this systematic review and meta-analysis.

## Author Contributions

**Conceptualization:** Martin Ackah.

**Data curation:** Martin Ackah, Louise Ameyaw, Mohammed Gazali Salifu.

**Formal analysis:** Martin Ackah, Mohammed Gazali Salifu.

**Investigation:** Martin Ackah, Louise Ameyaw, Mohammed Gazali Salifu, Cynthia OseiYeboah, Abena Serwaa Ampomaa Agyemang, Kow Acquaah, Yaa Boatema Koranteng, Asabea Opare-Appiah.

**Methodology:** Martin Ackah.

**Project administration:** Louise Ameyaw, Abena Serwaa Ampomaa Agyemang.

**Resources:** Martin Ackah, Louise Ameyaw, Mohammed Gazali Salifu, Cynthia OseiYeboah, Abena Serwaa Ampomaa Agyemang, Kow Acquaah, Yaa Boatema Koranteng.

**Software:** Martin Ackah.

**Supervision:** Martin Ackah, Louise Ameyaw.

**Validation:** Mohammed Gazali Salifu, Cynthia OseiYeboah.

**Visualization:** Martin Ackah, Louise Ameyaw.

**Writing – original draft:** Martin Ackah, Louise Ameyaw, Mohammed Gazali Salifu, Cynthia OseiYeboah, Abena Serwaa Ampomaa Agyemang, Kow Acquaah, Yaa Boatema Koranteng, Asabea Opare-Appiah.

**Writing – review & editing:** Martin Ackah, Louise Ameyaw, Mohammed Gazali Salifu, Cynthia OseiYeboah, Kow Acquaah, Yaa Boatema Koranteng, Asabea Opare-Appiah.

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
