## [Decision Letter · Decision Letter 0]

1 Mar 2022

PGPH-D-21-00952

Estimated Burden, and Associated Factors of Urinary Incontinence among Sub-Saharan African Women aged 15-100 years: a systematic review and meta-analysis.

Dear Dr. Ackah,

Thank you for submitting your manuscript to PLOS Global Public Health. After careful consideration, we feel that it has merit but does not fully meet PLOS Global Public Health’s publication criteria as it currently stands. Therefore, we invite you to submit a revised version of the manuscript that addresses the points raised during the review process.

We look forward to receiving your revised manuscript.

Kind regards,

Rajat Das Gupta, M.D.

Academic Editor

Journal Requirements:

1. Please amend your Financial Disclosure statement. If you did not receive any funding for this study, please simply state: “The authors received no specific funding for this work.”

Additional Editor Comments (if provided):

Reviewers' comments:

Reviewer's Responses to Questions

**Comments to the Author**

1. Does this manuscript meet PLOS Global Public Health’s publication criteria? Is the manuscript technically sound, and do the data support the conclusions? The manuscript must describe methodologically and ethically rigorous research with conclusions that are appropriately drawn based on the data presented.

Reviewer #1: Partly

Reviewer #2: Partly

2. Has the statistical analysis been performed appropriately and rigorously?

Reviewer #1: Yes

Reviewer #2: Yes

3. Have the authors made all data underlying the findings in their manuscript fully available (please refer to the Data Availability Statement at the start of the manuscript PDF file)?

Reviewer #1: Yes

Reviewer #2: Yes

4. Is the manuscript presented in an intelligible fashion and written in standard English?

Reviewer #1: No

Reviewer #2: No

5. Review Comments to the Author

Reviewer #1: I appreciate the effort of the authors. However, there are certain methodological concerns that should be addressed before acceptance. Please find my comments here:

1. Page 5: “Hospital and community based-studies had been conducted for UI in SSA countries.” - Please provide a reference for this statement.

2. Inclusion criteria: Some sentences are incomplete. For example: “Original observational studies published in English. An adult SSA woman [≥18 years].” - these two sentences are incomplete and grammatically incorrect.

3. Exclusion criteria: It started with an incomplete statement. For example: “Studies reporting animal studies, reviews, commentaries, letter to editors.”

4. Exclusion criteria: Mention about language as only English language has been considered.

5. Search: “The search was limited to January, 2000- September, 2021.” It means the search timeline was between January 2020 to September 2021 which is not correct I guess. Did the author consider articles published during this time? Or, is this the timeframe of searching the databases. If the authors considered the timeline from inception to September 2021, please mention that clearly.

6. Database: Google scholar is a search engine, not a database. Africa Journal Online is a local repository. Authors should search SCOPUS, Web of Science, EMBASE (at least two of these three) in addition to Medline.

7. A comprehensive search strategy for the Medline search should be provided. The search strategy for other databases should be provided as a supplementary file. As per the PRISMA 2020 guidelines, a search strategy for all the databases should be provided.

8. The current search strategy provided by the authors is revealing 618,343 articles (searched on 25 January 2022) from Medline/Pubmed. I don’t know how the authors have got only 720 articles from Medline/Pubmed. Can you please explain?

9. Selection process: “To ensure a rigorous review strategy, any duplicate articles were reviewed individually by two independent co-authors [MA, and KA,] in a double blinded process, and then rejected before selecting a unique collection of papers for this study” - What do you mean by “any duplicate article”? Two independent authors should screen all the articles.

10. The screening of articles is done in two phases. At first the title abstract screening, and then the full-text screening. Both the stages should be performed by two independent review authors independently. It should be mentioned properly.

11. Results: Study selection: Authors have removed 550 articles even before the screening! This is strange. They should mention the specific reason for the pre-screening exclusion. The authors should mention the causes of exclusion of 175 articles that underwent full-text screening.

12. Characteristics of the included studies and quality assessment: “Nigeria had the highest number of eligible studies [n=14], followed by Ethiopia [n=3], followed by Ghana and South Africa with two studies each.” - This calculation is showing 21 articles. Initially, the authors mentioned that they have included 25 articles.

13. Please add a paragraph stating the research implication. What are the recommendations for conducting further research based on the findings of this systematic review?

14. What are the policy implications of the findings?

15. The authors included articles published in English only? Isn’t it a limitation?

16. Overall, there should be thorough language editing.

Reviewer #2: REVIEWER’S COMMENTS FOR MANUSCRIPT NUMBER PGPH-D-21-00952

I would like to thank the authors for choosing an interesting topic which is also a neglected issue. The authors have put a great effort to it and tried to conduct a comprehensive review. However, I think there is still scope of improvement. My comment on this paper is listed below:

Introduction:

• The introduction needs reorganization. The first paragraph may mislead the readers regarding the target topic. I would suggest focusing on UI in the opening paragraph, keeping only one or two sentences regarding pelvic floor muscles and pelvic floor disorders

• It would be great if the authors could provide reference of some low-and-middle income countries also, which are comparable to the SSA countries

• The 2nd paragraph needs reorganization and resequencing of the sentences. Here, the authors provided definition and subtypes, followed by consequences. Then, all on a sudden, they provided US data, which seem irrelevant in the context

• The notion regarding limitation of underestimation of the studies conducted in SSA has not been references

• The rationale of doing a systematic review should be at the last paragraph

• The last para van go to conclusion

Research question

• Should be described in narrative instead of bullets

• This can also be incorporated in the last sentence of introduction

Methods

• It is great that the review is PROSPERO registered and it followed PRISMA guidelines

• The inclusion criteria and exclusion criteria are very poorly written. Many sentences are incomplete and need major revision

• The search strategy is explained well, but grammar checks

• The authors can add a supplementary table regarding Newcastle-Ottawa Quality Assessment findings

• For the data synthesis, I would suggest the authors to recheck the last sentence which reads “to evaluate the effects of probable factors influencing study heterogeneity, the meta-regression test was performed.” I reckon there is some problem in the sentence

Results:

• The authors documented I-square value of 91% but did not describe the heterogeneity well. This high heterogeneity of the studies puts the findings of meta-regression into question.

• I would suggest the authors to describe the heterogeneity among the studies more elaborately

• Secondly, I would suggest the authors to do a sub-group analysis with the studies that are more homogenous, and check whether it can replicate the overall result

• The authors could describe the findings from associated factors more elaborately

Discussion:

• The discussion should be rearranged according to the changes made in the result section

6. PLOS authors have the option to publish the peer review history of their article (what does this mean?). If published, this will include your full peer review and any attached files.

**Do you want your identity to be public for this peer review?** For information about this choice, including consent withdrawal, please see our Privacy Policy.

Reviewer #1: **Yes: **KM Saif-Ur-Rahman

Reviewer #2: **Yes: **Mohiuddin Ahsanul Kabir Chowdhury

---

## [Decision Letter · Decision Letter 1]

11 May 2022

Estimated Burden, and Associated Factors of Urinary Incontinence among Sub-Saharan African Women aged 15-100 years: a systematic review and meta-analysis.

PGPH-D-21-00952R1

Dear Martin Ackah,

We are pleased to inform you that your manuscript 'Estimated Burden, and Associated Factors of Urinary Incontinence among Sub-Saharan African Women aged 15-100 years: a systematic review and meta-analysis.' has been provisionally accepted for publication in PLOS Global Public Health.

Best regards,

Rajat Das Das Gupta, M.D.

Academic Editor

Reviewer Comments (if any, and for reference):

Reviewer's Responses to Questions

**Comments to the Author**

1. If the authors have adequately addressed your comments raised in a previous round of review and you feel that this manuscript is now acceptable for publication, you may indicate that here to bypass the “Comments to the Author” section, enter your conflict of interest statement in the “Confidential to Editor” section, and submit your "Accept" recommendation.

Reviewer #1: All comments have been addressed

Reviewer #2: All comments have been addressed

2. Does this manuscript meet PLOS Global Public Health’s publication criteria? Is the manuscript technically sound, and do the data support the conclusions? The manuscript must describe methodologically and ethically rigorous research with conclusions that are appropriately drawn based on the data presented.

Reviewer #1: Yes

Reviewer #2: Yes

3. Has the statistical analysis been performed appropriately and rigorously?

Reviewer #1: Yes

Reviewer #2: Yes

4. Have the authors made all data underlying the findings in their manuscript fully available (please refer to the Data Availability Statement at the start of the manuscript PDF file)?

Reviewer #1: Yes

Reviewer #2: Yes

5. Is the manuscript presented in an intelligible fashion and written in standard English?

Reviewer #1: Yes

Reviewer #2: Yes

6. Review Comments to the Author

Reviewer #1: Thanks for addressing the comments.

Reviewer #2: Well done

7. PLOS authors have the option to publish the peer review history of their article (what does this mean?). If published, this will include your full peer review and any attached files.

**Do you want your identity to be public for this peer review?** For information about this choice, including consent withdrawal, please see our Privacy Policy.

Reviewer #1: **Yes: **K.M. Saif-Ur-Rahman

Reviewer #2: **Yes: **Mohiuddin Ahsanul Kabir Chowdhury
